# Impact of Weight on Clinical Outcomes of Edoxaban Therapy in Atrial Fibrillation Patients Included in the ETNA-AF-Europe Registry

**DOI:** 10.3390/jcm10132879

**Published:** 2021-06-29

**Authors:** Giuseppe Boriani, Raffaele De Caterina, Marius Constantin Manu, José Souza, Ladislav Pecen, Paulus Kirchhof

**Affiliations:** 1Cardiology Division, Department of Biomedical, Metabolic and Neural Sciences, University of Modena and Reggio Emilia, Policlinico di Modena, 41100 Modena, Italy; 2Chair of Cardiology, Cardiology Division, Pisa University Hospital, University of Pisa, Via Paradisa 2, 56124 Pisa, Italy; raffaele.decaterina@unipi.it; 3Fondazione Villa Serena per la Ricerca, Città Sant’Angelo, 65013 Pescara, Italy; 4Daiichi Sankyo Europe GmbH, Zielstattstraße 48, 81379 Munich, Germany; Marius.Manu@daiichi-sankyo.eu (M.C.M.); Jose.Souza@daiichi-sankyo.eu (J.S.); 5Institute of Computer Science of the Czech Academy of Sciences, 18207 Prague, Czech Republic; Ladislav.Pecen@seznam.cz; 6Department of Cardiology, University Heart and Vascular Centre UKE Hamburg, 20246 Hamburg, Germany; p.kirchhof@uke.de; 7Institute of Cardiovascular Sciences, University of Birmingham, SWBH and UHB NHS Trusts, Birmingham B152TT, UK; 8The Atrial Fibrillation NETwork (AFNET), 48149 Münster, Germany

**Keywords:** atrial fibrillation, edoxaban, weight, registry, effectiveness, safety, non-vitamin K antagonist oral anticoagulants, stroke

## Abstract

Background: Extremes of body weight may alter exposure to non-vitamin K antagonist oral anticoagulants and thereby impact clinical outcomes. This ETNA-AF-Europe sub-analysis assessed 1-year outcomes in routine care patients with atrial fibrillation across a range of body weight groups treated with edoxaban. Methods: ETNA-AF-Europe is a multinational, multicentre, observational study conducted in 825 sites in 10 European countries. Overall, 1310, 5565, 4346 and 1446 enrolled patients were categorised into ≤60 kg, >60–≤80 kg (reference weight group), >80–≤100 kg and >100 kg groups. Results: Patients weighing ≤60 kg were older, more frail and had a higher CHA_2_DS_2_-VASc score vs. the other weight groups. The rates of stroke/systemic embolism, major bleeding and ICH were low at 1 year (0.82, 1.05 and 0.24%/year), with no significant differences among weight groups. The annualised event rates of all-cause death were 3.50%/year in the overall population. After adjustment for eGFR and CHA_2_DS_2_-VASc score, the risk of all-cause death was significantly higher in extreme weight groups vs. the reference group. Conclusions: Low rates of stroke and bleeding were reported with edoxaban, independent of weight. The risk of all-cause death was higher in extremes of weight vs. the reference group after adjustment for important risk modifiers, thus no obesity paradox was observed.

## 1. Introduction

Overweight and obesity are established risk factors of atrial fibrillation (AF) but the impact of underweight on AF has not been fully explored. The pathological mechanisms explaining the link between obesity and AF are complex and not fully understood; however, an increase in epicardial adipose tissue, enlargement of left atrium and diastolic dysfunction, have been associated with atrial remodeling leading to increased risk of AF in overweight/obese individuals [1]. 

At extremes of body weight, both thrombotic and haemorrhagic risks may be increased, and the pharmacokinetics of anticoagulant drugs may be altered, with a substantial risk of overdosing in underweight and under-dosing in obese individuals [2]. The complexity of the problem is enhanced by the limited amount of pharmacokinetic/pharmacodynamic studies performed in patients at extremes of body weight [3,4].

Non-vitamin K antagonist oral anticoagulants (NOACs) are the preferred treatment option for stroke prevention in patients with AF [5]. NOACs do not require routine blood testing for measuring coagulation because of the fewer drug interactions and a wide therapeutic window. Moreover, all four phase III randomised trials comparing NOACs vs. VKAs were performed without dose adjustments based on plasma level measurements [6]. Although routine monitoring of plasma levels of NOACs and subsequent dose adjustment is generally discouraged, data are emerging for the association between NOAC plasma levels and thrombotic and bleeding complications. One observational study (START) in patients with AF treated with NOACs (dabigatran, rivaroxaban, or apixaban) reported a relationship between low C-trough levels of NOACs and the occurrence of thrombotic events whereas bleeding complications were more frequent in those with higher C-peak levels of NOACs implicating the need for a more accurate assessment of NOAC doses [7,8]. Thus, the anticoagulant effect of NOACs has been observed to be related to their plasma concentration, which is further affected by body distribution volume. Therefore, extremes of body weight (such as underweight and obesity) may alter the exposure to NOACs and thereby impact their safety and efficacy [9,10,11]. It is still unclear if altered pharmacokinetic parameters may lead to an increased risk of bleeding in low body weight patients or an increased risk of stroke/systemic embolism in high-body weight patients [9,12].

Due to the concerns of decreased drug exposure and risk of under-dosing in the obese population, the International Society of Thrombosis and Haemostasis Scientific and Standardisation Committee (ISTH SSC) suggested in 2016 to avoid the use of NOACs in patients weighing >120 kg or with a BMI >40 kg/m^2^, but also provided no clear guidance on the use of NOACs in patients with low body weight due to the paucity of data [13]. 

This sub-group analysis of the ETNA-AF-Europe registry aimed to analyse the clinical outcomes in routine care patients treated with edoxaban across a range of body weight (>100 kg, ≤60 kg) as compared with the reference body weight group (>60–≤80 kg). In addition, as a secondary analysis, we also assessed the clinical outcomes in routine care patients treated with edoxaban across a range of BMI (≥35 kg/m^2^, <18.5 kg/m^2^) as compared with a normal BMI group (≥18.5–<25 kg/m^2^).

## 2. Materials and Methods

ETNA-AF-Europe (Clinicaltrials.gov: NCT02944019) is a multinational, multicentre, post-authorisation, observational study conducted in 825 sites in 10 European countries (Austria, Belgium, Germany, Ireland, Italy, The Netherlands, Portugal, Spain, Switzerland, and UK) [14,15].

The observational plan of this post-authorisation observational study was agreed in close collaboration with the European Medicines Agency (EMA) and has been reported previously [15]. The study was approved by the institutional review boards and Independent Ethics Committees for all participating centres in compliance with the Declaration of Helsinki and Guidelines for Good Pharmacoepidemiological Practice (GPP). All participants provided written informed consent.

One-year follow-up outcomes of the first 13,092 prospectively enrolled patients treated with edoxaban, have been published previously [16]. The present sub-analysis categorised patients into ≤60 kg, >60–≤80 kg (reference weight group), >80–≤100 kg and >100 kg groups. In the absence of universally defined categories of body weight, the limits were based on the threshold for dose adjustments of oral anticoagulants (60 kg) and on the basis of a grouping according to 20 kg ranges (60–80 and 80–100 kg). 

In addition, patients were also analysed according to their body mass index; patients were classified, as commonly done, as underweight (<18.5 kg/m^2^), normal weight (≥18.5–<25 kg/m^2^), overweight (≥25–<30 kg/m^2^), Class 1 obesity (≥30–<35 kg/m^2^) and Class 2 + 3 obesity (≥ 35 kg/m^2^).

Baseline characteristics are summarised descriptively as frequencies (*n* and percentage), mean value ± standard deviation (SD) rounded to integer, or median [interquartile range (IQR)]. Standardised differences of weight groups versus >60–≤80 kg group are reported for the baseline demographics and clinical characteristics.

There was no specific definition for frailty; it was left to the discretion of the physician to categorise a patient as frail (clinical evaluation).

Based on the reported events, annualised event rates (percentage per patient-year) are reported for the safety and effectiveness outcomes. Hazard ratios versus the reference group (>60–≤80 kg) are reported both for unadjusted data (univariate Cox proportional hazard model) and for data adjusted for eGFR and CHA_2_DS_2_-VASc score (multivariate Cox proportional hazard model). 

In the univariate analysis, age, eGFR and CHA_2_DS_2_-VASc were always amongst the top predictors for all analysed outcomes. Age is included in the CHA_2_DS_2_-VASc score therefore, eGFR and CHA_2_DS_2_-VASc were selected for adjustment.

All analyses were done using SAS version 9.4 for windows 10 (64-bit version). Univariate and multivariate Cox proportional hazard models were fitted by SAS proc PHREG.

## 3. Results

### 3.1. Baseline Characteristics

Overall, there were 1310, 5565, 4346 and 1446 patients in the ≤60 kg, >60–≤80 kg, >80–≤100 kg, and >100 kg weight groups, respectively (Table 1). As shown in Appendix A, there were too few patients in sub-groups of patients weighing <50 kg and >120 kg; therefore, ≤60 kg, >60–≤80 kg, >80–≤100 kg, and >100 kg weight groups were analysed. Baseline characteristics differed markedly among weight subgroups. Patients with lower body weight (≤60 kg) were older, more likely female, more frail, had a lower CrCl and a higher CHA_2_DS_2_-VASc score compared with reference weight group (>60–≤80 kg) and also in comparison with patients weighing >80–≤100 kg, and >100 kg (Table 1). As expected, a history of diabetes was more common in patients with higher body weights.

The proportion of patients with a history of stroke and major bleeding was higher in the lower weight category compared with reference weight group and patients weighing >80–≤100 kg, and >100 kg (Table 1). Paroxysmal AF was more common in lower body weight patients whereas a greater proportion of patients weighing >80–≤100 kg and >100 kg had persistent AF at baseline (Table 1). The proportions of patients meeting ≥1 edoxaban dose reduction criteria decreased with increasing weight, with 100% of lower weight patients meeting the criteria versus 2.4% in >100 kg weight category (Table 1).

Despite all patients meeting the dose reduction criteria in the lower weight group, only 62.7% received the recommended edoxaban 30 mg dose suggesting that the remaining 37.3% received higher than recommended doses at baseline (Table 1). A limited number of patients weighing >80–≤100 kg and >100 kg received lower than recommended doses (Table 1).

The large majority of patients ranged between body weight 65–90 kg (Figure 1), with the highest frequency of men weighing 80 kg and women weighing 70 kg (Appendix A). Differences in baseline characteristics followed a similar pattern when the patients were categorised according to their BMI (Appendix A).

### 3.2. Outcomes

#### 3.2.1. Stroke and Systemic Embolism

The risks of stroke/SEE and ischaemic stroke were low at 1 year 0.82%/year and 0.56%/year, respectively), with no differences in the risk among weight groups both in the unadjusted model and after adjustment for eGFR and CHA_2_DS_2_-VASc score (Figure 2 and Figure 3). In patients categorised by weight groups, the annualised event rates for stroke/SEE were 0.89%, 0.76% and 0.57% in the ≤60 kg, >80–≤100 kg and >100 kg groups versus 0.90% in the reference weight group, and for ischaemic stroke were 0.65%, 0.0.40%, 0.43% versus 0.65%, respectively.

#### 3.2.2. Bleeding

The event rates of major bleeding in the overall population were 1.05%/year and that of ICH were 0.24%/year. In patients categorised by weight groups, the annualised event rates for major bleeding were 1.30%, 0.91% and 0.43% in the ≤60 kg, >80–≤100 kg and >100 kg groups versus 1.28% in the reference weight group, and for ICH were 0.16%, 0.26%, 0.14% versus 0.24%, respectively. The analysis showed that the differences in risk according to weight groups were not significant except a lower risk of major bleeding and major or CRNMB bleeding in patients weighing >100 kg in unadjusted analysis however, it was no longer observed after adjustment for eGFR and CHA_2_DS_2_-VASc score (Figure 2 and Figure 3).

#### 3.2.3. Death Due to Any Cause and CV Deaths

The annualised event rates of death due to any cause and CV related deaths decreased with increasing weight (Figure 2 and Figure 3), with an annualised event rate of 1.63%/year for CV death and of 3.50%/year for death due to any cause in the overall population. In patients categorised by weight groups, the annualised event rates for all-cause death were 6.86%, 2.82% and 2.71% in the ≤60 kg, >80–≤100 kg and >100 kg groups versus 3.51% in the reference weight group.

In the unadjusted model, compared with the reference weight group (>60–≤80 kg), there was a significant increase in all-cause deaths in the lower weight group (≤60 kg: HR: 1.96, 1.52, 2.53; *p* < 0.0001), and a non-significant trend of decrease in death due to any cause with increasing weight (Figure 2). However, after adjustment for eGFR and CHA_2_DS_2_-VASc score, the risk of all-cause death was significantly higher in both the lower weight group and the groups of patients weighing >80–≤100 kg, and >100 kg in comparison with the reference weight group, thus depicting a U shaped pattern. The risk appeared to double in the ≥100 kg weight group compared with the reference weight group (≤60 kg: HR: 1.41, 95% CI: 1.08, 1.84; *p* = 0.0127; >80–≤100 kg: HR 1.29, 95% CI: 1.01, 1.65; *p* = 0.0438; ≥100 kg: HR: 2.04, 95% CI 1.35, 3.10; *p* = 0.0008) (Figure 3). 

In patients categorised by weight groups, the annualised event rates for CV death were 2.74%, 1.26% and 1.21% in the ≤60 kg, >80–≤100 kg and >100 kg groups versus 1.81% in the reference weight group. In the unadjusted model, compared with the reference weight group (>60–≤80 kg), the risk of CV-related deaths was significantly higher in the lower weight group (≤60 kg: HR: 1.52, 95% CI 1.03, 2.24; *p* = 0.0370) and had a non-significant trend for decrease with increasing weight (Figure 2). However, after adjustment for eGFR and CHA_2_DS_2_-VASc score, the risk of CV-related deaths did not different significantly according to weight categories (Figure 3).

#### 3.2.4. Additional Analyses

Appendix A include unadjusted and adjusted data for outcomes split by gender and weight categories, and these data show that the increase in the risk of mortality tended to be higher in women versus men in >80–≤100 kg and >100 kg weight groups.

The data for weight categories are further supplemented by data split by BMI categories, with some differences in the risks of outcomes observed in BMI categories versus the weight groups (Appendix A). After adjustment for eGFR and CHA_2_DS_2_-VASc, the risk of death and CV death was significantly increased in underweight patients as compared with patients with normal BMI (*p* = 0.0006 for death due to any cause and *p* = 0.0194 for CV death). Conversely, no significant differences in the risk of death due to any cause and CV deaths were observed with higher than normal BMI compared with the reference BMI category (≥18.5–<25 kg/m^2^). 

Appendix A show that large majority of patients ranged between 24–30 kg/m^2^, with the highest frequency of men and women in the 26 kg/m^2^ BMI category.

## 4. Discussion

This observational study showed that the risk of stroke and bleeding is low across a range of weight groups in patients with AF treated with edoxaban. Furthermore, while low body weight (≤60 kg) was associated with increased overall death, weight was not associated with different rates of stroke, major bleeding, or CV death in this population.

Historically, low body weight (<50 kg) has been an independent predictor of increased risk of ischaemic stroke, major bleeding, intracranial haemorrhage and death irrespective of the oral anticoagulation therapy [17]. Overall, patients with body weight <60 kg were under-represented in NOAC trials compared with those with body weight >100 kg. In the present ETNA-AF-Europe analysis, 10% of patients weighed ≤60 kg counting towards the existing evidence for the effectiveness and safety of NOACs in low body weight patients. In the individual NOAC trials, the efficacy and safety of the NOACs in patients with low body weight was consistent with the overall findings, warranting dose reduction in the case of edoxaban in participants weighing ≤60 kg and in case of apixaban when accompanied by at least one other clinical criterion [9]. A *post hoc* analysis of ARISTOTLE demonstrated that apixaban 5 mg BID (reduced to 2.5 mg BID if patients had ≥2 of the following criteria: age ≥80 years, body weight ≤60 kg, or a serum creatinine level ≥1.5 mg/dL) is efficacious and reduced major bleeding versus warfarin across the weight categories, including low- (≤60 kg) and high-weight patients (>120 kg) [18]. Of note, 1985 (10.9%) of the patients were in the low-weight group (≤60 kg).

Inappropriate dosing is not uncommon with NOACs. Use of off-label dosing regimen may be influenced by various factors including clinician perception of higher bleeding risk and lack of familiarity with dosing guidelines [19,20]. More commonly, inappropriate low dosing has been reported in studies exploring daily practice of NOACs [21,22]. Similar findings were observed in our analysis, with a limited number of patients weighing >80 kg received lower than the recommended dose. Such findings emphasise the need to explore strategies to prevent unintentional inappropriate dosing by prescribers.

Limited data obtained from trials and retrospective studies available thus far advocate the need for dedicated larger studies to address the use of NOACs in low weight patients. 

In the present ETNA-AF-Europe analysis, no significant differences in stroke-related outcomes were observed between body weight subgroups. These findings were suggestive of consistent effectiveness of edoxaban in a wide range of weight groups. There was a trend towards lower bleeding events with increasing weight in the unadjusted analysis. In particular, there was a significantly lower risk for major and clinically relevant non-major bleeding in patients >100 kg compared with the reference weight category. However, this difference was not noticeable after adjusting for eGFR and CHA_2_DS_2_-VASc score. Our findings are coherent with the outcomes observed in the sub-analysis of the ENGAGE AF-TIMI 48 randomised clinical trial that showed a similar risk of stroke/SE between edoxaban- and warfarin-treated patients across the three weight groups (low ≤55 kg, middle 79.8–84 kg and high ≥120 kg body weight) [3]. Major bleeding and net clinical outcome (composite of stroke, SEE, major bleeding, or death) were favourable with edoxaban versus warfarin especially in low body weight patients [3].

The evidence available so far suggests that despite the reduced risk of thromboembolic events with the use of oral anticoagulants, a high risk of CV-related deaths has been evident and requires the attention of physicians and policy makers [23]. The impact of associated comorbidities on mortality in AF patients, deserves to be tested outside of the randomised controlled trials setting. Findings from our analysis showed that the weight category deviating from the reference weight of 60–80 kg implied up to a 2-fold increase in the risk of all-cause death and numerical increases in CV death. In the unadjusted analysis, the risk of death due to any cause or CV deaths was significantly higher in the lower weight subgroup (≤60 kg) versus the reference weight group. Furthermore, the risk of all-cause death appeared to be higher in the lower weight group even after adjustment for eGFR and CHA_2_DS_2_-VASc score. There was a trend of decrease in the risk of death due to any cause and CV deaths in the higher weight groups in the unadjusted analysis. However, the risk appeared to increase with increasing weight after adjustment for eGFR and CHA_2_DS_2_-VASc score. Hence, no obesity paradox was observed in the high weight subgroups in the adjusted analysis. This is contrary to other studies in which obesity paradox was observed, potentially because of differences in the adopted adjustment for risk factors [24,25]. The vital impact of impaired renal function, as an independent variable, on outcomes in AF, as in many other diseases, explains these findings [26,27,28,29,30]. 

Interestingly, patients with lower weight in our analysis appeared to have protection from stroke and bleeding events after adjustment although they had a higher risk of death. These findings point towards optimisation of interventions with the goal to improve the management and outcomes of AF patients, according to the “ABC” approach (Atrial fibrillation Better Care [includes A (avoid stroke), B (better symptom control), and C (cardiovascular risk factors and comorbid conditions management)]) developed by European Society of Cardiology [5].

Our analysis was focused on weight categories, which in daily practice are more easily applicable as compared with BMI categories. Indeed, the practitioner usually takes into account body weight for adjusting the dose of NOACs, in combination with assessment of renal function. On the other hand, BMI assessment is largely used in epidemiological studies and is not regularly evaluated in daily practice for baseline patient assessments.

The association between BMI and clinical outcomes in patients with AF has been evaluated in various studies but the findings so far have been not concordant. Moreover, some of these studies had incomplete adjustment for confounding factors at baseline and the others were limited by small sample size and retrospective nature of the analyses. Higher BMI and waist circumference were associated with a favourable prognosis (adjusting for comorbidities and treatment allocation) in patients treated with oral anticoagulants [31,32]. In an analysis of patients treated with edoxaban and warfarin included in the ENGAGE AF-TIMI 48 trial, higher BMI (per 5 kg/m^2^ increase) was independently associated with lower risks of stroke/SEE and death, but with increased risk of major bleeding. The effects of edoxaban versus warfarin on stroke/SE, major bleeding, and net clinical outcome were similar across BMI categories ranging from 18.5 to >40 [33].

In a systematic review, an obesity paradox was observed for death due to any cause and CV-related deaths in the analysis of randomised NOAC trials; however, the analysis of the observational studies did not show this relationship. The analysis of the randomised trials also showed an obesity paradox for stroke/SEE and major bleeding, with a treatment effect favouring NOACs versus warfarin that was only significant for normal weight patients. No significant differences were observed between NOACs and warfarin in overweight and obese patients [24].

Some registry data also add to existing conflicting evidence from the randomised trials for the association between BMI and clinical outcomes in AF patients treated with oral anticoagulants. Findings from the DRESDEN registry evaluated the impact of BMI on clinical outcomes in daily care patients treated with NOACs for stroke prevention in AF [34]. There was no indication for the association between high BMI and decreased NOAC effectiveness or safety, which is in line with the obesity paradox that suggests a favourable effect of higher BMI on safety and effectiveness outcomes.

In the ORBIT-AF registry, all-cause death rates over the 2-year follow period decreased across increasing BMI categories, with highest rates in the normal weight group and lowest in the class III obesity group. However, it should be noted that obese patients were significantly younger than normal weight patients, and although age was included in the adjustment model, it may not have been possible to fully take the age differences into consideration [35]. Furthermore, BMI was noted to be inversely proportional to the risk for stroke/TIA/SEE in the unadjusted analysis; however, the association was no longer significant after adjustment for the risk factors.

Contrary to these findings, real word data from the prospective PREFER in AF and PREFER in AF PROLONGATION registries showed no obesity paradox for thromboembolic events in patients with AF [36]. Obese patients were at a higher risk of TEE, and oral anticoagulation was found to reduce the risk of events. Moreover, low body weight as well as obesity were also associated with increased bleeding.

Until recently, there has been a limited availability of robust clinical data to support definitive prescribing recommendations of NOACs in patients with BMI >40 kg/m^2^ or >120 kg total body weight. NOAC-specific pharmacokinetic variations have been observed in morbidly obese patients, with different NOACs showing differences in outcomes when used in patients with a high BMI [37,38,39]. However, data are also emerging on the use of NOACs in these patients suggesting that NOACs may be a safe and effective alternative to warfarin for prevention of stroke or SEE in morbidly obese patients [37,40,41]. Recent analyses of the four pivotal NOAC trials (RE-LY, ROCKET-AF, ARISTOTLE, and ENGAGE AF-TIMI 48) stratified by BMI showed preserved efficacy with NOACs compared with warfarin in obese patients, with a similar risk of major bleeding [42]. Although the data for class III obese patients (BMI ≥40 kg/m^2^) are limited, apixaban and edoxaban demonstrated similar efficacy and safety to warfarin in patients with BMI 40–50 kg/m^2^. In the ETNA-AF-Europe analysis, ~20% of patients ranged between 30–<35 kg/m^2^ and ~9% had a BMI ≥ 35 kg/m^2^ adding to the limited evidence in patients with a very high BMI.

We acknowledge the limitations of this analysis, which are typical of observational studies [43,44]. In this ETNA-AF-Europe sub-analysis, direct comparison of casual treatment effects was not possible as it did not involve randomised selection of therapy and only patients treated with edoxaban were eligible. The open-label nature of this observational study may have introduced bias due to knowledge about treatment. Although the study included both low and high weight patients, the data on extremes of body weight are limited, with only 10% of patients weighing ≤60 kg and ~11% weighing >100 kg. Furthermore, 90% of the patients included in our analysis are over 60 years of age. The extrapolation of these data for younger patients may not be possible because they may report somewhat different findings in comparison with older individuals.

## 5. Conclusions

In this ETNA-AF-Europe analysis, edoxaban seemed associated with a very low occurrence of stroke and bleeding, independently of the weight category in this wide European prospective observational study. 

A higher risk of death due to any cause was observed in lower and higher weight patients compared with the intermediate weight patients (reference weight category) after adjustment for important risk modifiers. This finding is in contrast with studies that described an “obesity paradox” for mortality. The data available thus far have also shown that the obesity paradox is not evident after statistical adjustments for associated comorbidities such as old age.

Findings from ETNA-AF-Europe suggest that patients with low body weight do well with a dose reduction. Moreover, the data also suggest that in overweight and obese patients, standard doses are adequate to achieve improved outcomes. To conclude, patients at extremes of body weight reported low rates of stroke and bleeding events with edoxaban within the current dosing guidelines. 

## Figures and Tables

**Figure 1 jcm-10-02879-f001:**
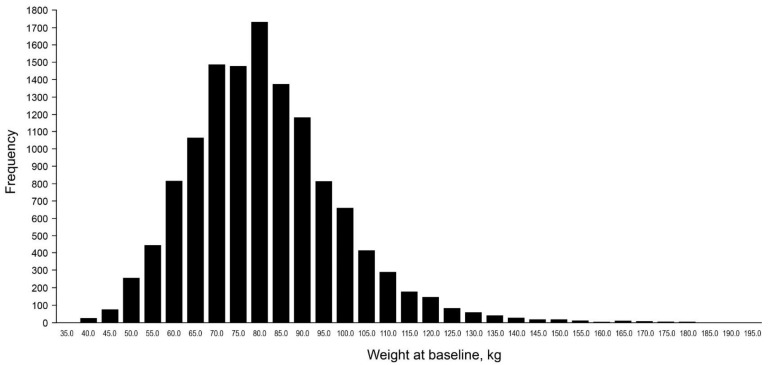
Frequency of patients across the range of body weight at baseline.

**Figure 2 jcm-10-02879-f002:**
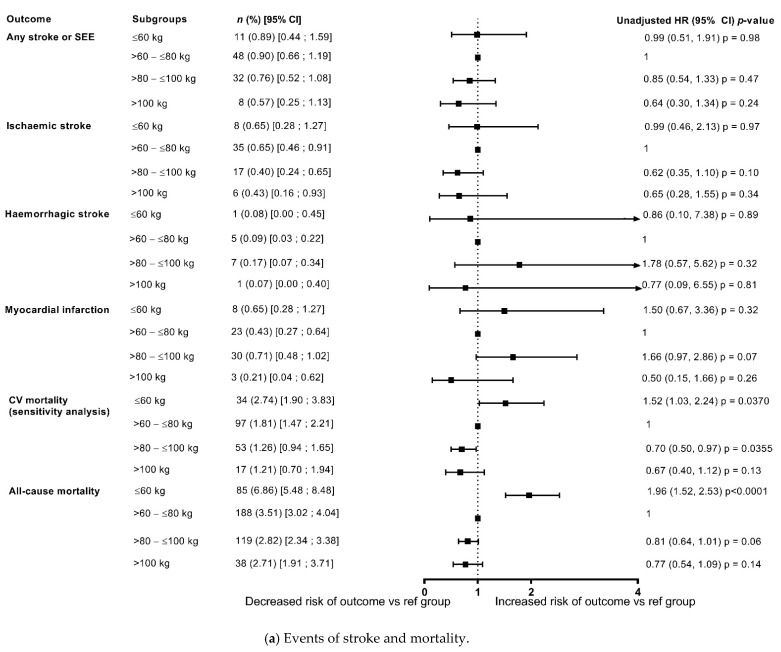
Outcomes at 1-year categorised by body weight (unadjusted data) compared with reference weight group (>60–≤80 kg). (**a**) Events of stroke and mortality; (**b**) Bleeding events. CRNM, clinically relevant non-major bleeding; GI, gastrointestinal; ICH, intracranial haemorrhage.

**Figure 3 jcm-10-02879-f003:**
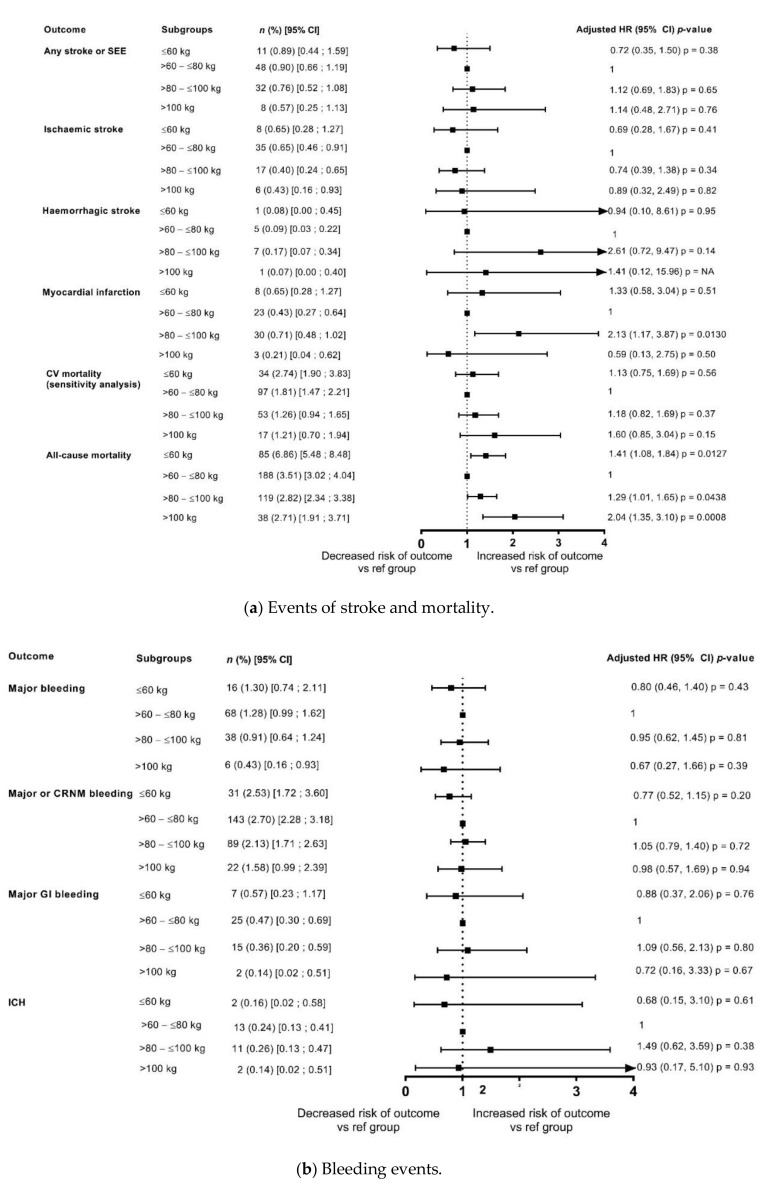
Outcomes at 1 year categorised by body weight (adjusted for eGFR and CHA_2_DS_2_-VASc score) compared with reference weight group (>60–≤80 kg). (**a**) Events of stroke and mortality; (**b**) Bleeding events. CRNM, clinically relevant non-major bleeding; GI, gastrointestinal; ICH, intracranial haemorrhage.

**Table 1 jcm-10-02879-t001:** Baseline demographics and clinical characteristics of patients included in ETNA-AF-Europe categorised by body weight.

	ETNA-AF-Europe Total	Body Weight [kg]						
		≤60 kg		>60–≤80 kg	>80–≤100 kg		>100 kg	
Patients, N (%)	13,092 (100%)	1310 (10.0%)	Standardized Difference to Weight >60–≤80 kg	5565 (42.5%)	4346 (33.2%)	Standardized Difference to Weight >60–≤80 kg	1446 (11.0%)	Standardized Difference to Weight ≤60 kg
Female, *n* (%)	5661 (43.2)	1145 (87.4)	0.83	2914 (52.4)	1107 (25.5)	−0.57	323 (22.4)	−0.65
Age, years mean ± SD median (Q1; Q3) <65 years 65–<75 years 75–<85 years ≥85 years	73.6 ± 9.46 75 (68; 80) 1994 (15.2) 4456 (34.0) 5268 (40.2) 1372 (10.5)	78.4 ± 8.64 79 (74; 84) 69 (5.3) 303 (23.1) 618 (47.2) 320 (24.4)	0.33	75.6 ± 8.62 77 (71; 81) 537 (9.6) 1685 (30.3) 2590 (46.5) 753 (13.5)	72.0 ± 9.02 73 (67; 79) 792 (18.2) 1707 (39.3) 1602 (36.9) 244 (5.6)	−0.41	66.6 ± 9.66 68 (60; 73) 541 (37.4) 601 (41.6) 289 (20.0) 14 (1.0)	−1.02
Body weight, kg min, max mean ± SD median (Q1; Q3) ≤60 kg 60–<80 kg 80–<100 kg ≥100 kg	38.0, 193.0 81.0 ± 17.29 80.0 (70.0; 90.0) 1310 (10.3) 5565 (43.9) 4346 (34.3) 1446 (11.4)	38.0, 60.0 55.0 ± 4.64 56.0 (52.0; 59.0) 1310 (100.0%) 0 (0.0) 0 (0.0) 0 (0.0)	−2.41	61.0, 80.0 72.0 ± 5.59 72.0 (68.0; 77.0) 0 (0.0) 5565 (100.0) 0 (0.0) 0 (0.0)	81.0, 100.0 89.3 ± 5.61 89.0 (85.0; 94.0) 0 (0.0) 0 (0.0) 4346 (100.0) 0 (0.0)	2.11	101.0, 193.0 113.8 ± 13.26 110.0 (105.0; 119.0) 0 (0.0) 0 (0.0) 0 (0.0) 1446 (100.0)	4.09
BMI, kg/m^2^ min, max mean ± SD median (Q1; Q3) <18.5 kg/m^2^ 18.5–<25 kg/m^2^ 25–<30 kg/m^2^ 30–<35 kg/m^2^ 35–<40 kg/m^2^ ≥40 kg/m^2^ ≥30 kg/m^2^	13.8, 68.6 28.1 ± 5.11 27.3 (24.7; 30.7) 115 (0.9) 3341 (26.7) 5377 (42.9) 2544 (20.3) 792 (6.3) 352 (2.8) 3688 (29.5)	13.8, 28.5 21.8 ±2.33 21.8 (20.2;23.4) 110 (8.5) 1066 (82.7) 113 (8.8) 0 (0.0) 0 (0.0) 0 (0.0) 0 (0.0)	−1.61	15.0, 39.4 26.0 ± 2.72 25.7 (24.2; 27.6) 5 (0.1) 2083 (37.8) 2934 (53.3) 459 (8.3) 24 (0.4) 0 (0.0) 483 (8.8)	21.0, 46.3 29.8 ± 3.34 29.4 (27.5;31.6) 0 (0.0) 191 (4.4) 2249 (52.3) 1519 (35.3) 304 (7.1) 36 (0.8) 1859 (43.2)	1.27	24.0, 68.6 36.6 ± 5.35 35.5 (33.0; 39.4) 0 (0.0) 1 (0.1) 81 (5.7) 566 (39.6) 464 (32.5) 316 (22.1) 1346 (94.3)	3.08
CrCl (recalc.), mL/min, mean ± SD median (Q1;Q3)	74.3 ± 30.42 69.8 (53.0; 89.6)	48.7 ± 17.26 47.0 (36.8; 58.7)	−0.77	64.1 ± 20.47 62.6 (49.3; 76.4)	82.2 ± 25.54 80.6 (64.4; 97.7)	0.79	115.0 ± 38.26 110.3 (88.0; 137.0)	2.03
CHADS_2_ (recalc.) mean ± SD median (Q1;Q3)	1.7 ± 1.07 2 (1; 2)	1.9 ± 1.09 2 (1; 2)	0.08	1.8 ± 1.08 2 (1; 2)	1.7 ± 1.07 2 (1; 2)	−0.09	1.6 ± 0.95 1 (1; 2)	−0.22
CHA_2_DS_2_-VASc (recalc.) mean ± SD median (Q1;Q3) 0 1 2 3 ≥4	3.1 ± 1.40 3 (2; 4) 290 (2.2) 1324 (10.1) 2802 (21.4) 3768 (28.8) 4908 (37.5)	3.8 ± 1.26 4 (3; 4) 2 (0.2) 29 (2.2) 149 (11.4) 356 (27.2) 774 (59.1)	0.37	3.3 ± 1.36 3 (2; 4) 91 (1.6) 378 (6.8) 1047 (18.8) 1638 (29.4) 2411 (43.3)	2.9 ± 1.37 3 (2; 4) 119 (2.7) 574 (13.2) 1101 (25.3) 1270 (29.2) 1282 (29.5)	−0.33	2.5 ± 1.30 2 (2; 3) 62 (4.3) 283 (19.6) 413 (28.6) 387 (26.8) 301 (20.8)	−0.60
mod. HAS-BLED (recalc.) mean ± SD median (Q1; Q3) <3 ≥3	2.5 ± 1.10 2 (2;3) 6711 (51.3) 6381 (48.7)	2.6 ± 1.05 3 (2;3) 648 (49.5) 662 (50.5)	0.02	2.6 ± 1.09 3 (2;3) 2748 (49.4) 2817 (50.6)	2.5 ± 1.12 3 (2;3) 2164 (49.8) 2182 (50.2)	−0.03	2.3 ± 1.09 2 (2;3) 856 (59.2) 590 (40.8)	−0.24
Frailty *, *n* (%)								
Yes	1392 (10.6)	341 (26.0)	0.37	663 (11.9)	293 (6.7)	−0.18	66 (4.6)	−0.27
No	10,820 (82.7)	878 (67.0)	−0.34	4554 (81.8)	3778 (87.0)	0.14	1270 (87.9)	0.17
Not known	878 (6.7)	91 (6.9)	0.03	348 (6.3)	274 (6.3)	0.002	109 (7.5)	0.05
Medical history, *n* (%)								
Hypertension	10,088 (77.1)	940 (71.8)	−0.10	4230 (76.0)	3443 (79.2)	0.08	1206 (83.4)	0.18
Diabetes	2879 (22.0)	199 (15.2)	−0.09	1041 (18.7)	1080 (24.9)	0.15	487 (33.7)	0.35
Dys-/hyperlipidaemia	5626 (43.0)	487 (37.2)	−0.12	2401 (43.1)	1980 (45.6)	0.05	610 (42.2)	−0.02
Coronary heart disease	2738 (20.9)	220 (16.8)	−0.09	1123 (20.2)	1031 (23.7)	0.09	308 (21.3)	0.03
Peripheral artery disease	437 (3.3)	44 (3.4)	0.0009	186 (3.3)	152 (3.5)	0.009	46 (3.2)	−0.009
Congestive heart failure	1850 (14.2)	189 (14.4)	0.0006	801 (14.4)	601 (13.9)	−0.003	225 (15.6)	0.06
Myocardial infarction	560 (4.3)	57 (4.4)	0.02	223 (4.0)	211 (4.9)	0.04	57 (3.9)	−0.003
TIA	448 (3.4)	52 (4.0)	0.02	202 (3.6)	136 (3.1)	−0.03	34 (2.4)	−0.08
History of stroke & ICH, *n* (%)								
Ischaemic stroke	778 (5.9)	102 (7.8)	0.06	350 (6.3)	237 (5.5)	−0.04	48 (3.3)	−0.14
Intracranial haemorrhage	62 (0.5)	8 (0.6)	0.02	26 (0.5)	22 (0.5)	0.006	4 (0.3)	−0.03
History of bleeding, *n* (%)	424 (3.2)	59 (4.5)	0.07	178 (3.2)	133 (3.1)	−0.008	44 (3.0)	−0.009
Major	129 (1.0)	23 (1.8)	0.07	51 (0.9)	39 (0.9)	−0.002	13 (0.9)	−0.002
Major or CRNM	270 (2.1)	44 (3.4)	0.08	114 (2.0)	77 (1.8)	−0.02	29 (2.0)	−0.003
Current AF type, *n* (%)								
Paroxysmal	7039 (53.9)	754 (57.7)	0.05	3062 (55.1)	2233 (51.4)	−0.07	713 (49.4)	−0.11
Persistent	3159 (24.2)	263 (20.1)	−0.06	1243 (22.4)	1160 (26.7)	0.10	424 (29.4)	0.16
Long-standing persistent & Permanent	2864 (21.9)	289 (22.1)	−0.009	1248 (22.5)	948 (21.8)	−0.01	305 (21.2)	−0.03
Patients fulfilling ≥1 dose adjustment criteria,^†^ *n* (%)	3106 (23.7)	1310 (100.0)	2.50	1352 (24.3)	408 (9.4)	−0.41	35 (2.4)	−0.68
Edoxaban dose at baseline, *n* (%) 60 mg 30 mg	9991 (76.3) 3101 (23.7)	488 (37.3) 822 (62.7)	−0.85 0.85	4238 (76.2) 1327 (23.8)	3637 (83.7) 709 (16.3)	0.19 −0.19	1278 (88.4) 168 (11.6)	0.32 −0.32

* There was no specific definition for frailty; it was left to the discretion of the physician to categorise a patient as frail. ^†^ Patients with one or more of the following clinical factors were dose reduced: moderate or severe renal impairment (CrCl 15–50 mL/min), low body weight <60 kg, or concomitant use of certain P-glycoprotein inhibitors (These criteria apply to Europe and may differ outside of the European Union). AF, atrial fibrillation; BMI, body mass index; COPD, chronic obstructive pulmonary disease; CrCl, creatinine clearance; CRNM, clinically relevant non-major; CV, cardiovascular; ICH, intracranial haemorrhage; OD, once daily; Q1, lower quartile; Q3, upper quartile; SD, standard deviation; TIA, transient ischaemic attack.

## Data Availability

The data underlying this article are available in the article and in its online Appendix A.

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
