# Peer review of "Impact of Weight on Clinical Outcomes of Edoxaban Therapy in Atrial Fibrillation Patients Included in the ETNA-AF-Europe Registry"

_jcm, 2021, doi:10.3390/jcm10132879_

Round 1

Reviewer 1 Report

The authors provide an interesting

manuscript providing contrasting evidence about the suggested obesity paradox.

some minor points should be addressed: the authors might explain in more detail how body mass and body weight may contribute to AF.

There are recent data in edoxaban

substudies focussing on different outcome

in AF types. Did AF phenotype like

paroxysmal AF had an effect on outcome? 
This should be mentioned in the text.

Reviewer 2 Report

This manuscript evaluating the impact of weight on clinical outcomes of edoxaban (a non-vitamin K antagonist oral anticoagulant) therapy in atrial fibrillation patients presents valuable knowledge for increasing confidence and clinical management in patient specific care. Atrial fibrillation is already the most commonly occurring arrhythmia and growing, while obesity is also a growing concern. The conclusions from this study that currently applied dosing strategies based on patient body mass, provides greater confidence in the application of anti-coagulation therapy to treat co-morbidities driven by atrial fibrillation. Overall, the generation of data and the conclusions drawn are clear, however, it would be beneficial for the manuscript to have a more clear description of methodologies and procedures used to evaluate the dataset prior to publication.

Concerns

Overall, how was the multivariable model constructed? Was a univariable analysis performed to identify statistically significant variables as a screen to then populate the multivariable model? Or was another basis used for populating the variables in the final multivariable model?

What software and statistical model/analysis (such as a Multivariable Cox proportional hazards analysis) was used for generation of the results.

Adjusted variables eGFR and CHA2DS2-VASc score were identified as important risk modifiers within the abstract, but no other rationalization was provided for why these two variables were used. While CHA2DS2-VASc scoring is more clear in the assessment for edoxaban performance, eGFR would benefit from a more clear description for its inclusion as an adjusted variable. Does edoxaban have renal clearance providing additional rationalization for adjusting to eGFR, or is this strictly there for its significance as a risk modifier?

Reviewer 3 Report

The study by Boriani et al. is a reaffirmation that DOAC therapy is safe and effective at extremes of weight when dosed as recommended.  I have a couple of general comments about this study.

First, the authors might comment on the fact that these drugs have a wide therapeutic window (vis a vis a drug like warfarin), especially when looked at in the short (or immediate) term given the differences in peak and trough levels, but there is evidence that in the longer term, individuals with “average” higher peaks or “average” lower troughs are ones that might get into trouble (i.e., bleeding vs strokes).

The many different outcomes are a little confusing for the reader.  CV related death and all cause mortality are definitely important but may have many possible causes unrelated to anticoagulant therapy. The outcomes that one can relate directly to the anticoagulant or its concentration are SSE , ischemic stroke, hemorrhagic stroke, major and CRNM bleeds.  I suggest emphasizing these outcomes which might directly reflect drug concentration and the other outcomes secondarily.    

For the US audience, it should be noted that the EMA recommends a dose reduction not only for CrCL of 15 – 50 ml/min, but also for weight < 60kg.  In the US, it is only for the lower CrCL.

~2/3rds of the lower weight group were appropriately dosed at the lower dose, but ~1/3rd was not.  It would be interesting to know how that 1/3rd did. Did they experience a higher bleed rate? One would also have to factor in the CrCL in these patients. The same might be said for the high weight group; about 12% received a lower dose (presumably for low CrCL?).

These issues should be discussed briefly in the Discussion and also addressed in the limitations of the study.  I think the authors should modify their conclusions to recognize that the best or most appropriate dose for low weight individuals is not definitively known, but the study suggests that such individuals do well with a dose reduction.  This is important information for the US audience.  Regarding the high weight group, there is little evidence to suggest that such individuals require a higher than standard dose to achieve good outcomes.  The bottom line, as the authors say, is that patients at extremes of weight do well with current dosing guidelines.

Page 2 line 48-51:  I agree that the anticoagulant effect of the DOACs is closely related to their plasma concentration.  What is not fully clear is whether the clinical effect of the DOACs is related to their plasma concentration.

Page 11, lines231-232:  Regarding the lower dose in the high weight group (~12%), perhaps they all had CrCL of < 50??  Do the authors know?
